# Measurement of Off-Flavoring Volatile Compounds and Microbial Load as a Probable Marker for Keeping Quality of Pasteurized Milk

**Anjum Rashid [1], Imran Javed [2], Barbara Rasco [1], Shyam Sablani [1], Muhammad Ayaz [2], Muhammad A. Ali [3], Muhammad Abdullah [2], Muhammad Imran [4], Tanweer Aslam Gondal [5], Muhammad Inam Afzal [6], Muhammad Atif [7], Bahare Salehi [8,*], Célia F. Rodrigues [9], Javad Sharifi-Rad [10,*] and Natália Martins [11,12,*]**

[1] School of Food Sciences, Washington State University, Pullman, WA 99164-5110, USA; anjofst@gmail.com (A.R.); rasco@wsu.edu (B.R.); ssablani@wsu.edu (S.S.)

[2] Department of Dairy Technology, University of Veterinary and Animal Sciences, Lahore 5400, Pakistan; imranjaved@uvas.edu.pk (I.J.); Muhammad.ayaz@uvas.edu.pk (M.A.); drmabdullah.uvas@yahoo.com (M.A.)

[3] Department of Food Science and Human Nutrition, University of Veterinary and Animal Sciences, Lahore 5400, Pakistan; asif.ali@uvas.edu.pk

[4] University Institute of Diet and Nutritional Sciences, The University of Lahore, Lahore 54000, Pakistan; mic_1661@yahoo.com

[5] School of Exercise and Nutrition, Deakin University, Victoria 3217, Australia; tgondal@deakin.edu.au

[6] Department of Biosciences, COMSATS University Islamabad, Tarlai Kalan, Islamabad 46000, Pakistan; dr-miafzal@hotmail.com

[7] Department of Clinical Laboratory Sciences, College of Applied Medical Sciences, Jouf University, Sakaka 72341, Saudi Arabia; aatif03@gmail.com

[8] Student Research Committee, Bam University of Medical Sciences, Bam 44340847, Iran

[9] LEPABE—Department of Chemical Engineering, Faculty of Engineering, University of Porto, Rua Dr. Roberto Frias, s/n, 4200-465 Porto, Portugal; c.fortunae@gmail.com

[10] Food Safety Research Center (salt), Semnan University of Medical Sciences, Semnan 3519899951, Iran

[11] Faculty of Medicine, University of Porto, Alameda Prof. Hernâni Monteiro, 4200-319 Porto, Portugal

[12] Institute for Research and Innovation in Health (i3S), University of Porto, 4200-135 Porto, Portugal

[*] Correspondence: bahar.salehi007@gmail.com (B.S.); javad.sharifirad@gmail.com (J.S.-R.); ncmartins@med.up.pt (N.M.); Tel.: +98-34-4434-1120 (B.S.); +351-22-551-2100 (N.M.); +98-21-8820-0104 (J.S.-R.)

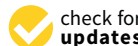

**Featured Application: The current study was conducted to identify and measure the off-flavoring volatile compounds and microbial load of pasteurized fluid milk, stored at different temperatures, as a possible indicator of its keeping quality.**

**Abstract:** (1) Background: Multiple attempts have been conducted to correlate milk keeping quality with chemical, physical or bacteriological parameters. These methods only measure the chemical changes in milk produced by bacteria. Headspace solid-phase micro-extraction (HS-SPME) is an economic and recent method used to measure both volatile compounds and microbial load in milk, also allowing to keep the quality of the milk product. (2) Methods: The present study was conducted to identify and measure the off-flavoring volatile compounds through gas chromatography coupled with flame ionization detector (GC-FID) and the microbial load of pasteurized fluid milk stored at different temperatures, as a possible indicator of its keeping quality. (3) Results: The highest results were obtained to acetone, followed by butanone, pentanal and ethanol. These mean values were significantly enhanced from the 0 to 19th day of storage, at 10 °C. At day 19th, the minimum score for aroma, flavor and overall acceptability were also recorded as $4.33 \pm 0.17$, $4.02 \pm 0.06$, $4.00 \pm 0.04$, respectively. Likewise, maximum values for standard plate count ($\text{Log}_{10}$ CFU $15.54 \pm$

0.40 mL$^{-1}$) and total psychotroph count (Log$_{10}$ CFU 11.67 $\pm$ 0.30mL$^{-1}$) were reported at 10 °C and 4 °C. (4) Conclusion: HS-SPME/GC-FID methodology revealed to be very sensitive and capable to be applied in volatile compounds quantification in pasteurized milk produced during the storage period at different temperatures.

**Keywords:** pasteurized milk; microbial load; off-flavoring volatile compounds; storage stability

## 1. Introduction

It has been demonstrated that it is more difficult to objectively quantify milk spoilage than organoleptically. Some types of milk, with a standard plate count of $10^6$ CFU/mL, are perfectly palatable, whereas others have become unpalatable. Further, it is even more difficult to estimate "keeping quality", the number of days between manufacture and spoilage [1].

Keeping quality is essentially determined by the rate of production of off-flavors and off-tastes. As such, it is only indirectly related to the bacterial quality of milk. Thus, a bacterial strain which produces more off-flavors will obviously cause spoilage at a lower count than one which produces lesser off-flavors [2]. In addition, off-flavor production may also vary with temperature, storage time and substrate availability, which itself may be produced by other bacteria [3].

Pasteurized milk-induced spoilage causes significant economic losses to food industry. Dogan and Boor [4] reported that the presence of large populations of *Pseudomonas* spp. generally resulted in shorter shelf-lives for pasteurized milk than if products were contaminated with other types of organisms [4]. Thus, a mixed culture of bacteria may produce more off-flavors than those containing individual strains.

Many attempts have been made to correlate the keeping quality of milk with chemical, physical or bacteriological parameters. The most promising methods have been impedance measurement [5], and acetaldehyde estimation in milk headspace [6]. Nonetheless, all of these methods measure chemical changes produced by bacteria rather than their actual numbers and as such might be expected to correlate better with organoleptic quality than would standard plate count or psychotropic count [7].

Volatile compounds in milk have been extensively studied using different extraction techniques including static headspace, purge and trap, and solvent-assisted flavor evaporation [8–10]. The headspace solid-phase microextraction (HS-SPME) has been developed. This technique is able to substantially reduce analysis time and sample manipulation steps [11]. Indeed, HS-SPME has been widely used to extract volatile components from dairy foods, such as fluid processed milk, cheese, milk powder and milk chocolate [12]. On the other hand, many studies have reported the off-flavor development in milk at a given storage time, usually at the end of the product shelf-life. However, there is limited information in literature regarding changes in milk flavor compounds, as affected by storage time and distinct temperatures. Thus, the present study aims to identify and to quantify volatile compounds in pasteurized milk, stored at different temperatures and specific time periods, for possibly correlated changes in volatile compounds to sensory and microbiological attributes for keeping milk quality.

## 2. Materials and Methods

### 2.1. Samples and Storage Conditions

Commercially available pasteurized milk with 3.25% standardized fat content with the same date of production was purchased from Walmart Superstore of Darigold company brand (Inc. Seattle, Washington, DC, USA). Samples were transferred, under sterilized conditions, to a 250 mL plastic vessel. A total of 30 samples were stored at 4, 7 and 10 °C for 19 days with a time interval of 0, 1, 5, 9, 12, 14, 16, 17, 18, and 19 days, for further analysis.

## 2.2. Microbiological Analysis

To determine total microbial load, total plate count was assessed, using dilution standard method 42-11. The samples were also scrutinized for total coli form count to infer about sewerage water contamination and adulteration in milk used for fermented dairy products production, adapting the approved method 42-15. Media was changed to Violet Red Bile without autoclaving, and the remaining methodology was the same as applied for total plate count.

For the psychotropic count, plate count skim milk agar (Merck) was used to determine total psychotropic bacteria count (TPC) and it was applied by a standard method 42-45 [13]. The samples were inoculated by three successive dilutions with sterile saline solution, in three replicates.

## 2.3. Extraction of Volatile Compounds and Solid Phase Micro Extraction

The headspace (HS) solid phase micro extraction (SPME) technique was adopted for absorption of milk off-flavoring volatile compounds. In this technique, the milk samples were agitated with the addition of salt and the temperature was maintained at $18 \pm 1\ ^\circ C$ to overcome the remains of flavor compounds. The salt was added to improve the release of volatile compounds that ultimately increased the absorption efficiency of SPME fiber.

For this study, 10 mL milk sample was taken in a 25 mL flask equipped with cap and Teflon-faced silicone rubber septa (Supelco, Co., Bellefonte, PA, USA). A magnetic stir plate (model PC-220, Corning, NY, USA) was used for stirring the flask containing the milk sample and salt at 1100 rpm 20 min. The volatiles were preequilibrated by adding an internal standard n-tridecane @ 0.5 mg/10 mL of sample after 18 min of stirring. Volatile compounds were extracted using HS-SPME, according to a previously described method in Dalgado et al. [14]. Three fibers from a flavor collection kit were examined: 75 μm carboxen/poly-dimethylsiloxane (CAR/PDMS), 100 μm poly-dimethylsiloxane (PDMS) and 65 μm poly-dimethylsiloxane/divinilyebene (PDMS/DVB), founded upon its affinity with milk spoilage volatiles and with the symmetry of chromatographic peaks (time exposure: 1 h). When the fiber was chosen, four exposure times (30, 45, 60, and 90 min) were considered, to improve the extraction of volatile compounds.

## 2.4. Identification and Quantification Volatile by GC-FID

The extracted/adsorbed volatiles into SPME fiber were thermally desorbed into injector port of Gas chromatographs (GC) Varian 3400 (Varian, Inc., Walnut Creek, CA, SUA) that was connected with a flame ionization detector (FID). The common conditions for GC-FID assembly were: the carrier gas was Helium, a DB-1 column (30 m × 0.32 mm i.d., 0.25 μm film thickness, J&W Scientific, Folsom, CA, USA), the injector temperature was 200 and detector 250 $^\circ C$ according to the method in Cramer et al. [15]. At the start the column remained at 33 $^\circ C$ for 5 min then increased at the rate of 2 $^\circ C$/min up to 50 $^\circ C$. Finally, it was adjusted at 225 $^\circ C$ by increasing at the rate of 5 $^\circ C$/min. A preliminary study was conducted for the first identification of off-flavoring volatile compounds by using GC-MS under the same conditions (but with a column length of 60 m), by matching their spectra with those present in Wiley-NBS library mass spectral data.

The same volatiles were purchased as standard compounds that were first identified at GC-MS in the preliminary study. The compounds Acetaldehyde, Acetone, Butanal, Butanone, Ethanol, Hex-Propanone and Hexanon were purchased from Chem Service (West Chester, PA, USA). The compounds Propanol, Eth-Butanol, Hexanal, 2 Meth-1-Pro-ol, Heptanal, 3Meth-1Butanol, Trans-2-Hexenal, Octanal were purchased from Sigma-Aldrich Corp. St. Louis, MO, USA. The compounds 2,3-Butanediol, Hex-Propanone, 1-Hexanol and 1-Pentanol were purchased from purchased from Alltech (Deerfield, IL, USA). These standard compounds were run on GC-FID and got a (Figure 1) of peaks of different volatiles according to their retention times. After that, the milk samples were run on GC-FID and identification and quantification of volatiles were achieved using detector response factors established from standard compounds and their retention time.

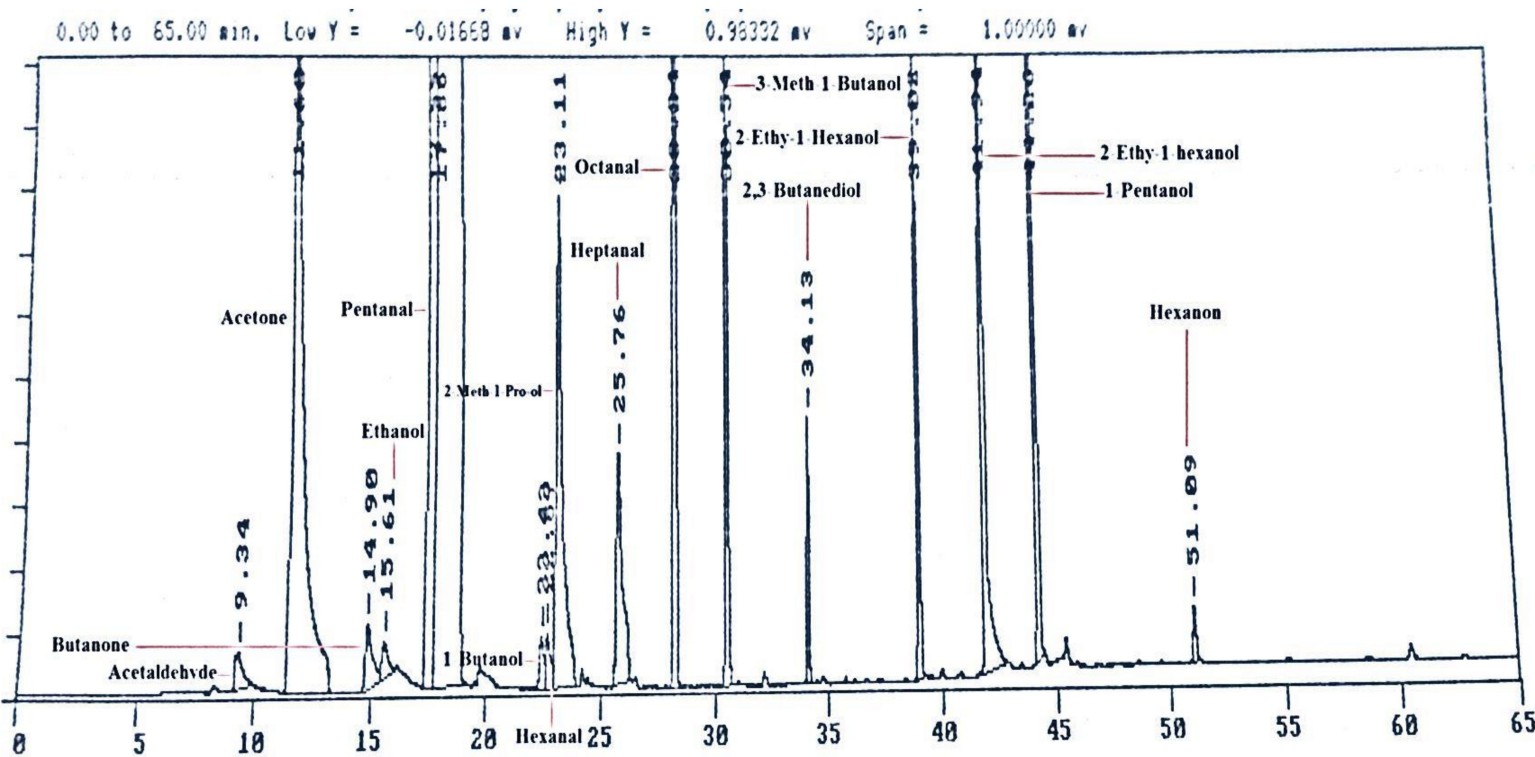

**Figure 1.** Standards volatile compounds run on gas chromatography coupled with flame ionization detector (GC-FID).

## 2.5. Sensory Analysis

Sensory stability of the samples was estimated by a trained panel, consisting of 10 members. Panelists were asked to evaluate the sensory attributes of aroma, flavor and overall acceptability by assigning a liking score on a 9-point hedonic by Meilgaard et al. [16].

## 2.6. Statistical Analysis

Data was analyzed under Complete of Randomized Design (CRD) through two-way analysis of variance (ANOVA) technique. Means separation was made using Duncan's Multiple Range Test. All statistical analysis was carried out using Mini-tab software package (SPSS-18, Armonk, NY, USA) [17].

## 3. Results and Discussion

### 3.1. Optimization of SPME Experimental Conditions

Various fiber coatings and exposure times were explored prior to analysis of volatile compositions of pasteurized milk. PDMS-coated fiber originated lower area counts of individual volatile compounds than that of PDMS/DVB or CAR/PDMS coatings (Figure 1). Moreover, PDMS/DVB and CAR/PDMS fibers revealed, correspondingly, 14 and 21 peaks on their respective chromatograms and showed adsorbed distinctive compounds. Consistent with the manufacturer, CAR/PDMS coating is more suitable for low molecular weight compounds detection (e.g., 2-butanone or acetone), while PDMS/DVB absorbs higher molecular weight compounds (e.g., nonanal or decanal) [18]. In the present work, the lowest molecular weight compound detected was 3-methylbutanal, and the highest one was heptanal. PDMS/DVB coated fiber proved to be superior for analysis of volatiles in pasteurized milk samples.

Regarding the SPME fiber exposure time to the headspace of the milk sample, it was noticed that, after 30 and 45 min, most of the volatile compounds were detected with PDMS/DVB fiber. Other compounds, for example heptanal, were only detectable after 60 min-exposure and 90 min-exposure time was suitable for SPME analysis of a milk sample, as the area count of most of the compounds (e.g., heptanal, 3-heptanone, 2-heptanone, and 2-pentylfuran) revealed small changes. Because the area count of hexanal or 1-octen-3-ol remained affected by exposure time after 90 min, this time was prudently controlled throughout the experiment in order to restrain variability among replications.

### 3.2. Quantification of Volatiles in Pasteurized Milk

Volatile compounds were produced in pasteurized milk as a result of storage time and temperature. A total of 20 volatile compounds were quantified in milk samples, ranging from $0.01 \pm 0.01$ µg/L to $139.6 \pm 6.7$ µg/L, which agrees with previous reports [9,19]. Freshly pasteurized milk (Table 1, day 0) at 4 °C resulted in the measurement of 6 compounds, including ketones (2), alcohols (2) and aldehydes (2), which were increased to 12 compounds, with 3 ketones, 7 alcohols and 2 aldehydes, at the end of storage (day 19). A comparable volatile profile was described by Marsili [20] for cow's milk with low fat content, at day 0. Carbonyl compounds are secondary products of lipid photo-oxidation, as product of light exposure and a microbial metabolite [21]. Of the above compounds, acetone was measured the compound with the maximum amount of $75.6 \pm 1.13$ µg/L, at 0 day, and $93.3 \pm 2.14$ µg/L at day 19. Butanone, ethanol and pentanal were also quantified compounds, which increased proportionally with storage time at 4 °C.

A similar volatile profile was detected for milk stored at 7 °C (Table 2). At day 0, the same 6 compounds were measured, but at the end of storage, the total number of volatile compounds increased to 14, including ketones (4), aldehydes (4) and alcohols (6). The acetone amount was $73.5 \pm 1.12$ µg/L at day 0 and $99.6 \pm 2.17$ µg/L at day 19, being slightly higher than at 4 °C. Two new compounds were detected at this temperature: acetaldehyde at day 17 and 3-Meth-1-Butanol at day 18, which was likely photo-oxidation of milk. This deterioration in packaging materials arises in milk in

the presence of a photosensitizer (e.g., riboflavin), which excites oxygen to its singlet state, a requisite for oxygen to react with the unsaturated lipid moiety, producing hydroperoxide [22].

The number of volatile compounds in pasteurized milk stored at 10 °C, at day 0, was almost similar to that at 4 °C and 7 °C, but high at end of storage. A total of 20 different compounds were observed, including aldehydes (5), ketones (4) and alcohols (11) (Table 3). At day 0, acetone (72.8 $\pm$ 1.32 µg/L) was the highest quantified compound, followed by butanone (7.56 $\pm$ 0.03 µg/L), pentanal (1.13 $\pm$ 0.02 µg/L) and ethanol (0.04 $\pm$ 0.01 µg/L). These compounds slightly increased at the 19[th] day of storage, being respectively, 139.6 $\pm$ 6.7, 15.4 $\pm$ 2.20, 11.2 $\pm$ 2.06 and 5.29 $\pm$ 1.17. These results are in close agreement with that obtained by Karatapanis et al. [23] and a similar profile of off-flavoring compounds was detected in pasteurized milk by Vazquez-Landaverde et al. [24]. Analyzing the trend of volatile compound production, mostly all compounds increased in quantity as the storage interval increased, but some decreased as the storage period increased, at all studied temperatures. At the same time, the rate of increase was higher at 10 °C. Adequate reproducibility was achieved for most compounds (relative standard deviation < 15%).

### 3.3. Microbial Count and Spoilage Indication

Standard plate count (SPC) and TPC of pasteurized milk was significantly influenced, while total coliform count (TCC) was insignificantly affected by both storage period and temperature, and even their interaction ($p < 0.05$). At 0 day, the mean values of TCC, SPC and TPC were, respectively, ($Log_{10}$ CFU 0.83 $\pm$ 0.14 mL$^{-1}$, 0.83 $\pm$ 0.14 mL$^{-1}$ and 0.83 $\pm$ 0.14 mL$^{-1}$) at 4 °C; ($Log_{10}$ CFU 3.14 $\pm$ 0.03 mL$^{-1}$, 3.14 $\pm$ 0.03 mL$^{-1}$ and 3.14 $\pm$ 0.03 mL$^{-1}$) at 7 °C; and ($Log_{10}$ CFU 2.24 $\pm$ 0.02 mL$^{-1}$, 2.24 $\pm$ 0.02 mL$^{-1}$ and 2.24 $\pm$ 0.02 mL$^{-1}$) at 10 °C.

**Table 1.** Means values and standard deviation of volatile compounds (µg/L) of pasteurized milk stored at 4 °C.

| Volatile Compounds | Storage Days | | | | | | | | | |
|---|---|---|---|---|---|---|---|---|---|---|
| | 0 | 1 | 5 | 9 | 12 | 14 | 16 | 17 | 18 | 19 |
| Acetone | 75.6 ± 1.13 | 77.4 ± 2.65 | 78.7 ± 1.53 | 82.6 ± 2.08 | 79.5 ± 3.01 | 84.9 ± 1.27 | 85.6 ± 2.05 | 87.4 ± 1.14 | 93.2 ± 1.07 | 93.3 ± 2.14 |
| Butanone | 8.06 ± 0 | 8.36 ± 0 | 8.68 ± 0.01 | 8.89 ± 1.04 | 9.09 ± 0 | 9.31 ± 1.02 | 9.42 ± 0.02 | 9.64 ± 2.06 | 9.86 ± 1.52 | 10.23 ± 2 |
| Ethanol | 0.01 ± 0.01 | 0.16 ± 0.1 | 1.07 ± 0.4 | 1.23 ± 0.45 | 1.67 ± 0.66 | 1.52 ± 0.52 | 1.54 ± 1.03 | 1.77 ± 1.24 | 1.83 ± 1.48 | 1.99 ± 1 |
| Pentanal | 1.21 ± 0.04 | 1.33 ± 0.06 | 1.89 ± 0.08 | 2.57 ± 0.92 | 3.15 ± 0.54 | 3.94 ± 0.73 | 4.82 ± 0.08 | 5.40 ± 0.65 | 6.15 ± 1.08 | 7.83 ± 1.14 |
| 1-Butanol | 0.02 ± 0 | 0.01 ± 0 | 0.01 ± 0 | 0.02 ± 0 | 0.03 ± 0.02 | 0.02 ± 0 | 0.03 ± 0.02 | 0.07 ± 0.01 | 0.13 ± 0.01 | 0.41 ± 0.02 |
| 3-Meth-1-Butanol | ND | ND | ND | ND | ND | ND | ND | ND | 0.02 ± 0.01 | 0.09 ± 0.01 |
| Heptanal | 0.01 ± 0 | 0.02 ± 02 | 0.02 ± 0.02 | 0.03 ± 0.01 | 0.07 ± 0.04 | 0.06 ± 0.02 | 0.13 ± 0.05 | 0.27 ± 0.10 | 0.45 ± 0.13 | 0.957 ± 0 |
| 2,3-Butanediol | ND | ND | ND | ND | ND | ND | ND | ND | 0.016 ± 0 | 0.017 ± 0 |
| 1-Hexanol | ND | ND | ND | ND | ND | ND | ND | ND | 0.002 ± 0 | 0.002 ± 0 |
| 2-Eth-1-Hexanol | ND | ND | ND | ND | ND | ND | ND | ND | 0.002 ± 0 | 0.003 ± 0 |
| 1-Pentanol | ND | ND | 0.05 | ND | ND | ND | ND | 0.59 ± 0.13 | 0.84 ± 0.2 | 0.95 ± 0.01 |
| Hexanone | ND | ND | ND | ND | ND | ND | ND | 0.065 ± 0 | 0.077 ± 0 | 0.088 ± 0 |

ND, not determined.

**Table 2.** Means values and standard deviation of volatile compounds (µg/L) of pasteurized milk stored at 7 °C.

| Volatile Compounds | Storage Days | | | | | | | | | |
|---|---|---|---|---|---|---|---|---|---|---|
| | 0 | 1 | 5 | 9 | 12 | 14 | 16 | 17 | 18 | 19 |
| Acetaldehyde | ND | ND | ND | ND | ND | ND | ND | 4.53 ± 0.64 | 8.34 ± 0 | 16.3 ± 0.70 |
| Acetone | 73.5 ± 1.12 | 74.3 ± 1.05 | 76.8 ± 2.03 | 85.5 ± 2.18 | 87.5 ± 3.01 | 88.7 ± 1.20 | 91.5 ± 2.05 | 93.6 ± 1.47 | 96.5 ± 2.05 | 99.6 ± 2.17 |
| Butanone | 7.06 ± 0.02 | 8.07 ± 0.02 | 9.10 ± 0.02 | 9.12 ± 0.01 | 10.3 ± 0.02 | 10.9 ± 0.10 | 11.4 ± 0.01 | 10.2 ± 0.02 | 11.3 ± 0.4 | 12.9 ± 1.20 |
| Ethanol | 0.03 ± 0. | 0.21 ± 0.2 | 0.57 ± 0.5 | 0.73 ± 0.45 | 0.97 ± 0.3 | 0.94 ± 0.02 | 1.64 ± 1.53 | 1.92 ± 0.01 | 2.11 ± 0.12 | 4.29 ± 0.12 |
| Pentanal | 1.17 ± 0.03 | 1.58 ± 0.05 | 2.05 ± 0.09 | 2.87 ± 0.12 | 3.37 ± 0.64 | 3.94 ± 0.73 | 5.12 ± 0.25 | 6.56 ± 0.72 | 7.91 ± 0.14 | 9.02 ± 1.03 |
| 1-Butanol | 0.02 | 0.03 ± 0 | 0.08 ± 0 | 0.12 ± 0.02 | 0.13 ± 0.03 | 0.22 ± 0.06 | 0.25 ± 0.02 | 0.57 ± 0.01 | 0.73 ± 0.01 | 0.91 ± 0.02 |
| Hexanal | ND | ND | ND | ND | ND | ND | ND | 0.012 ± 0 | 0.02 ± 0 | 0.05 ± 0.01 |
| 2-Meth-1-Pro-ol | ND | ND | ND | ND | ND | ND | ND | 0.000 | 0.06 ± 0.01 | 0.16 ± 0.05 |
| Heptanal | 0.02 ± 0 | 0.01 ± 0.01 | 0.01 ± 0.02 | 0.04 ± 0.03 | 0.09 ± 0.05 | 0.12 ± 0.06 | 0.16 ± 0.05 | 0.36 ± 0.14 | 0.91 ± 0.11 | 1.64 ± 0.32 |
| Octanal | ND | ND | ND | ND | ND | ND | ND | 0.000 | 0.02 ± 0 | 0.08 ± 0 |
| 3-Meth-1-Butanol | ND | ND | ND | ND | ND | ND | ND | 0.000 | 0.01 ± 0 | 0.06 ± 0.02 |
| 2,3-Butanediol | ND | ND | ND | ND | ND | ND | ND | 0.089 ± 0 | 0.09 ± 0.01 | 0.22 ± 0.04 |
| 2-Ethy-1-hexanol | ND | ND | ND | ND | ND | 0.001 ± 0 | 0.002 ± 0 | 0.002 ± 0 | 0.002 ± 0 | 0.01 ± 0 |
| 1-Pentanol | ND | ND | ND | ND | ND | 1.70 ± 0.76 | 0.96 ± 0.10 | 0.73 ± 0.46 | 0.63 ± 0.14 | 0.57 ± 0.77 |
| Hexanon | ND | ND | ND | ND | ND | 0.065 ± 0 | 0.07 ± 0 | 0.09 ± 0.05 | 0.09 ± 0.02 | 0.06 ± 0.04 |

ND, not determined.

**Table 3.** Means values and standard deviation of volatile compounds (µg/L) of pasteurized milk stored at 10 °C.

| Volatile Compounds | Storage Days | | | | | | | | | |
|---|---|---|---|---|---|---|---|---|---|---|
| | 0 | 1 | 5 | 9 | 12 | 14 | 16 | 17 | 18 | 19 |
| Acetaldehyde | ND | ND | ND | ND | ND | ND | 8.38 ± 3.02 | 8.62 ± 2.38 | 9.10 ± 0.83 | 9.99 ± 0.81 |
| Acetone | 72.8 ± 1.32 | 74.8 ± 1.25 | 79.8 ± 2.33 | 88.6 ± 2.38 | 90.4 ± 2.41 | 92.7 ± 1.28 | 97.3 ± 2.65 | 101.6 ± 2.4 | 109.5 ± 3.5 | 139.6 ± 6.7 |
| Butanal | ND | ND | ND | ND | ND | ND | ND | 0.31 ± 0.44 | 1.31 ± 0.59 | 1.63 ± 0.93 |
| Butanone | 7.56 ± 0.03 | 8.17 ± 0.05 | 9.19 ± 0.05 | 9.56 ± 0.04 | 10.9 ± 0.10 | 12.1 ± 0.10 | 12.3 ± 0.01 | 13.7 ± 0.12 | 13.3 ± 0.6 | 15.4 ± 2.20 |
| Ethanol | 0.04 ± 0.01 | 0.20 ± 0.02 | 0.87 ± 0.4 | 1.23 ± 0.85 | 1.43 ± 0.36 | 1.94 ± 0.12 | 2.34 ± 1.03 | 2.82 ± 0.06 | 3.71 ± 0.52 | 5.29 ± 1.17 |
| Pentanal | 1.13 ± 0.02 | 1.47 ± 0.04 | 2.65 ± 0.19 | 3.78 ± 0.12 | 3.37 ± 0.64 | 3.96 ± 0.5 | 5.72 ± 0.45 | 7.36 ± 0.65 | 8.92 ± 0.84 | 11.2 ± 2.06 |
| Propanol | ND | ND | ND | ND | ND | ND | 0.10 ± 0.02 | 0.14 ± 0.05 | 0.15 ± 0 | 0.16 ± 0 |
| Eth-Butanol | ND | ND | ND | ND | ND | ND | ND | 0.013 ± | 0.013 ± 0 | 0.014 ± 0 |
| Hexanal | ND | ND | ND | ND | ND | ND | 0.10 ± 0.01 | 0.10 ± 0 | 0.11 ± 0.01 | 0.12 ± 0.05 |
| 2-Meth-1-Pro-ol | ND | ND | ND | ND | ND | ND | 0.21 ± 0.13 | 0.54 ± 0.28 | 0.55 ± 0 | 0.56 ± 0.18 |
| Heptanal | 0.02 | 0.04 ± 0 | 0.18 ± 0.01 | 0.21 ± 0.03 | 0.43 ± 0.05 | 0.63 ± 0.04 | 0.55 ± 0.02 | 0.78 ± 0.08 | 1.07 ± 0.29 | 1.51 ± 0.43 |
| 3Meth-1Butanol | ND | ND | ND | 0.011 ± 0 | 0.01 ± 0 | 0.01 ± 0 | 0.02 ± 0 | 0.04 ± 0.19 | 0.09 ± 0.08 | 0.23 ± 0.11 |
| Trans-2-Hexenal | ND | ND | ND | 0.015 ± 0 | 0.015 ± 0 | 0.016 ± 0 | 0.02 ± 0 | 0.03 ± 0.01 | 0.05 ± 0.02 | 0.06 ± 0.02 |
| Octanal | ND | ND | ND | ND | ND | ND | 0.67 ± 0.01 | 0.68 ± 0.04 | 0.69 ± 0.03 | 0.71 ± 0.02 |
| 2,3-Butanediol | ND | ND | ND | ND | ND | ND | ND | 0.27 ± 0.03 | 0.16 ± 0.2 | 0.07 ± 0.01 |
| Hex- Propanone | ND | ND | ND | ND | ND | ND | ND | 0.07 ± 0 | 0.07 ± 0 | 0.06 ± 0 |
| 1-Hexanol | ND | ND | ND | ND | ND | ND | 0.002 ± 0 | 0.003 ± 0 | 0.003 ± 0 | 0.003 ± 0 |
| 1-Pentanol | ND | ND | ND | ND | ND | ND | 0.94 ± | 6.46 ± 1.68 | 23.73 ± 0.1 | 24.17 ± 0.1 |
| Hexanon | ND | ND | ND | ND | ND | ND | 0.13 ± 0.02 | 0.12 ± 0.02 | 0.05 ± 0.01 | 0.04 ± 0 |

ND, not determined.

At the 19th day of storage, TCC, SPC and TPC were increased to ($Log_{10}$ CFU $2.33 \pm 0.03$ mL$^{-1}$, $4.08 \pm 0.05$ mL$^{-1}$ and $5.36 \pm 0.17$ mL$^{-1}$) at 4 °C, ($Log_{10}$ CFU $7.90 \pm 0.11$ mL$^{-1}$, $12.13 \pm 0.19$ mL$^{-1}$ and $14.33 \pm 0.31$ mL$^{-1}$) at 7 °C, and ($Log_{10}$ CFU $13.892$ mL$^{-1}$, $11.896$ mL$^{-1}$ and $7.670$ mL$^{-1}$) at 10 °C, respectively (Table 4). To highlight that at 10 °C, *Enterobacteriaceae* and Gram-positive bacteria revealed to assume a greater importance in the pasteurized milk spoilage. The shelf-life values found at 4 and 7 °C were revealed to be longer when compared to values reported in under-developed countries. For pasteurized milk, stored at 7 °C, Shipe et al. [25] reported an average keeping quality of 13 days, and at 4 °C, of 18 to 20 days, while Martins et al. [26] reported 9, 12-13, and 15-17 days, respectively, for a short, medium and long shelf-life.

*3.4. Sensory Evaluation*

The results of sensory evaluation are shown in Table 5. In general, off-flavors increased more rapidly in milk stored at 10 °C than at 4 °C and 7 °C. After 14 days-storage, the overall acceptability score of milk stored at 4 °C was $8.00 \pm 0.29$, followed by at 7 °C ($7.50 \pm 0.00$) and 10 °C ($7.00 \pm 0.00$). Similarly, the score of taste and aroma at day 14 was ($7.50 \pm 0.00$, $7.17 \pm 0.60$), ($7.00 \pm 0.00$, $6.67 \pm 0.33$) and ($6.67 \pm 0.33$, $6.33 \pm 0.17$) at 4 °C, 7 °C and 10 °C, respectively. The overall acceptability of milk samples, after 19 days of storage, at 4 °C was (score $6.00 \pm 0.00$) going to spoil, and at 7 °C and 10 °C were badly spoiled, with scores of respectively, $5.17 \pm 0.17$ and $4.00 \pm 0.00$. Overall, the obtained sensory results agree with those of Lloyd et al. [27], who reported that storage-induced changes in sensory characteristics, volatiles and primary amines depend on both differences in lactose content and the applied heat processing, as well as with the results obtained by Li et al. [28], who reported that aldehyde and ketone compounds may play a major role in controlling the oxidized flavor, which was influenced by pre-heating as well as concentration and drying during milk powder production. The standard plate count (SPC, $Log_{10}$) was ($2.91 \pm 0.03$ CFU/mL, $5.77 \pm 0$ CFU/mL and $14.88 \pm 0.03$ CFU/mL), at day 14, and ($5.78 \pm 0.01$ CFU/mL, $17.25 \pm 0.03$ CFU/mL and $21.89 \pm 0.01$ CFU/mL), at day 19, respectively, at 4 °C, 7 °C and 10 °C. This has previously been noted by Labuza [29] and Maxcy and Wallen [30], who reported that the most reliable method to judge fresh milk quality is sensory evaluation.

**Table 4.** Means values and standard deviation of microbial count (Log$_{10}$ CFU/mL) of pasteurized milk stored at different temperatures.

| | Temp. | \multicolumn Storage Days | | | | | | | | | | Mean |
|---|---|---|---|---|---|---|---|---|---|---|---|---|
| | | 0 | 1 | 5 | 9 | 12 | 14 | 16 | 17 | 18 | 19 | |
| **TCC** | **4 °C** | 0.83 ± 0.14l | 0.83 ± 0.14l | 0.83 ± 0.15l | 0.96 ± 0.14l | 1.46 ± 0.07jkl | 1.75 ± 0.20h-k | 2.08 ± 0.07g-j | 2.16 ± 0.07ghi | 2.20 ± 0.12ghi | 2.33 ± 0.03gh | 1.54 ± 0.12C |
| | **7 °C** | 0.83 ± 0.14l | 0.83 ± 0.14l | 0.83 ± 0.14l | 1.29 ± 0.10kl | 1.79 ± 0.05h-k | 2.30 ± 0.06ghi | 2.56 ± 0.05fg | 3.04 ± 0.03ef | 3.63 ± 0.17de | 4.08 ± 0.05cd | 2.12 ± 0.21B |
| | **10 °C** | 0.83 ± 0.14l | 0.83 ± 0.13l | 0.96 ± 0.16l | 1.64 ± 0.09ijk | 2.47 ± 0.20fg | 3.56 ± 0.22de | 4.17 ± 0.01cd | 4.53 ± 0.01bc | 4.96 ± 0.01ab | 5.36 ± 0.17a | 2.93 ± 0.32A |
| | **Mean** | 0.83 ± 0.07G | 0.83 ± 0.07G | 0.87 ± 0.08G | 1.30 ± 0.11F | 1.91 ± 0.16E | 2.54 ± 0.28D | 2.94 ± 0.32C | 3.24 ± 0.35C | 3.60 ± 0.40B | 3.93 ± 0.44A | |
| **SPC** | **4 °C** | 3.14 ± 0.03n | 3.14 ± 0.03n | 3.17 ± 0.04n | 3.27 ± 0.06n | 3.72 ± 0.04mn | 4.41 ± 0.21lm | 5.46 ± 0.24jk | 6.92 ± 0.16gh | 7.14 ± 0.10gh | 7.90 ± 0.11g | 4.83 ± 0.33C |
| | **7 °C** | 3.14 ± 0.02n | 3.14 ± 0.07n | 3.22 ± 0.06n | 3.64 ± 0.03n | 4.80 ± 0.15ijk | 5.77 ± 0.12ij | 7.74 ± 0.17g | 9.87 ± 0.11f | 11.64 ± 0.31de | 12.13 ± 0.19d | 6.51 ± 0.63B |
| | **10 °C** | 3.14 ± 0.02n | 3.16 ± 0.02n | 3.38 ± 0.05n | 4.55 ± 0.05lm | 6.65 ± 0.33hi | 10.86 ± 0.20e | 12.67 ± 0.24c | 14.23 ± 0.34b | 15.54 ± 0.40a | 14.33 ± 0.31b | 8.85 ± 0.91A |
| | **Mean** | 3.14 ± 0.01G | 3.15 ± 0.02G | 3.25 ± 0.04G | 3.82 ± 0.19F | 5.06 ± 0.44E | 7.01 ± 0.99D | 8.62 ± 1.07C | 10.34 ± 1.07B | 11.44 ± 1.22A | 11.45 ± 0.95A | |
| **TPC** | **4 °C** | 2.24 ± 0.02o | 2.26 ± 0.03o | 2.61 ± 0.04o | 3.34 ± 0.03ij | 4.74 ± 0.02g | 5.86 ± 0.04e | 6.93 ± 0.05d | 7.76 ± 0.09c | 9.13 ± 0.10b | 11.67 ± 0.30a | 5.65 ± 0.57A |
| | **7 °C** | 2.24 ± 0.03o | 2.23 ± 0.01o | 2.32 ± 0.00o | 2.68 ± 0.05no | 2.86 ± 0.05klm | 3.44 ± 0.03i | 4.12 ± 0.05h | 4.95 ± 0.06fg | 5.27 ± 0.06f | 6.20 ± 0.12e | 3.63 ± 0.25B |
| | **10 °C** | 2.24 ± 0.02o | 2.25 ± 0.01o | 2.25 ± 0.03o | 2.29 ± 0.05no | 2.53 ± 0.05mn | 2.75 ± 0.03lm | 2.97 ± 0.06jkl | 3.22 ± 0.05ijk | 3.50 ± 0.01i | 3.99 ± 0.06h | 2.80 ± 0.11C |
| | **Mean** | 2.24 ± 0.01H | 2.25 ± 0.01H | 2.39 ± 0.06H | 2.77 ± 0.16G | 3.38 ± 0.34F | 4.02 ± 0.47E | 4.67 ± 0.59D | 5.31 ± 0.66C | 5.96 ± 0.83B | 7.29 ± 1.14A | |

SPC, standard plate count; TCC, total coliform count; TPC, total psychotropic bacteria count.

**Table 5.** Means values of score of sensory attributes of pasteurized milk stored at different temperatures.

| Parameter | Temp. | \multicolumn Storage Days | | | | | | | | | | Mean |
|---|---|---|---|---|---|---|---|---|---|---|---|---|
| | | 0 | 1 | 5 | 9 | 12 | 14 | 16 | 17 | 18 | 19 | |
| **Aroma** | **4 °C** | 9.00 ± 0.00 | 9.00 ± 0.00 | 9.00 ± 0.00 | 8.33 ± 0.33 | 8.00 ± 0.58 | 7.17 ± 0.60 | 7.00 ± 0.00 | 6.83 ± 0.17 | 6.33 ± 0.17 | 5.67 ± 0.44 | 7.63 ± 0.23A |
| | **7 °C** | 9.00 ± 0.00 | 9.00 ± 0.00 | 9.00 ± 0.00 | 8.33 ± 0.33 | 7.67 ± 0.17 | 6.67 ± 0.33 | 6.33 ± 0.17 | 6.33 ± 0.17 | 5.83 ± 0.17 | 5.17 ± 0.17 | 7.33 ± 0.26B |
| | **10 °C** | 9.00 ± 0.00 | 9.00 ± 0.00 | 8.33 ± 0.33 | 8.00 ± 0.58 | 7.00 ± 0.58 | 6.33 ± 0.17 | 6.17 ± 0.17 | 6.17 ± 0.17 | 5.67 ± 0.17 | 4.33 ± 0.17 | 7.00 ± 0.28C |
| | **Mean** | 9.00 ± 0.00A | 9.00 ± 0.00A | 8.78 ± 0.15AB | 8.22 ± 0.22BC | 7.56 ± 0.28C | 6.72 ± 0.24D | 6.50 ± 0.14DE | 6.44 ± 0.13DE | 5.94 ± 0.13E | 5.06 ± 0.24F | |
| **Taste** | **4 °C** | 9.00 ± 0.00 | 9.00 ± 0.00 | 9.00 ± 0.00 | 8.00 ± 0.50 | 8.00 ± 0.00 | 7.50 ± 0.00 | 7.00 ± 0.29 | 7.00 ± 0.58 | 6.50 ± 0.29 | 6.00 ± 0.29 | 7.70 ± 0.21A |
| | **7 °C** | 9.00 ± 0.00 | 9.00 ± 0.00 | 8.50 ± 0.00 | 8.33 ± 0.17 | 7.67 ± 0.33 | 7.00 ± 0.00 | 6.33 ± 0.33 | 6.00 ± 0.00 | 6.00 ± 0.29 | 5.00 ± 0.00 | 7.28 ± 0.25B |
| | **10 °C** | 9.00 ± 0.00 | 9.00 ± 0.00 | 8.00 ± 0.29 | 7.33 ± 0.33 | 7.00 ± 0.58 | 6.67 ± 0.33 | 5.50 ± 0.50 | 5.67 ± 0.33 | 5.00 ± 0.29 | 4.00 ± 0.06 | 6.72 ± 0.31C |
| | **Mean** | 9.00 ± 0.00A | 9.00 ± 0.00A | 8.50 ± 0.17AB | 7.89 ± 0.23BC | 7.56 ± 0.24CD | 7.06 ± 0.15D | 6.28 ± 0.29E | 6.22 ± 0.28E | 5.83 ± 0.26E | 5.00 ± 0.30F | |
| **OAA** | **4 °C** | 9.00 ± 0.00 | 9.00 ± 0.00 | 9.00 ± 0.00 | 9.00 ± 0.00 | 8.33 ± 0.17 | 8.00 ± 0.29 | 7.50 ± 0.29 | 7.33 ± 0.33 | 7.00 ± 0.58 | 6.00 ± 0.00 | 8.02 ± 0.20A |
| | **7 °C** | 9.00 ± 0.00 | 9.00 ± 0.00 | 8.50 ± 0.00 | 8.50 ± 0.29 | 7.83 ± 0.17 | 7.50 ± 0.00 | 7.00 ± 0.58 | 6.83 ± 0.44 | 6.17 ± 0.33 | 5.17 ± 0.17 | 7.55 ± 0.24B |
| | **10 °C** | 9.00 ± 0.00 | 9.00 ± 0.00 | 8.00 ± 0.58 | 8.17 ± 0.17 | 7.50 ± 0.29 | 7.00 ± 0.29 | 6.00 ± 0.58 | 5.83 ± 0.17 | 5.67 ± 0.17 | 4.02 ± 0.04 | 7.02 ± 0.30C |
| | **Mean** | 9.00 ± 0.00A | 9.00 ± 0.00A | 8.50 ± 0.22AB | 8.56 ± 0.15AB | 7.89 ± 0.16BC | 7.50 ± 0.19CD | 6.83 ± 0.33DE | 6.67 ± 0.28E | 6.28 ± 0.28E | 5.06 ± 0.29F | |

OAA, overall acceptability.

*3.5. Correlation between Sensory, Microbiological and Volatile Compounds*

Figures 2–10 signposted a linear rise in TCC, SPC and TPC, at all temperatures, with a maximum increase in SPC in the last storage days, mainly at 10 °C. This can be associated with the average acceptability scores, which slightly declined during the early days of milk storage. Though, at 7 °C and 10 °C, flavor rapidly deteriorated and milk samples were considered as unacceptable after days 16 and 12, respectively. Milk samples kept at 10 °C were excluded by the panel after days 12 to 14. The main milk defects described included acid, sour and rancid for samples kept at 10 °C, and bitter, rancid and slightly acidic for samples stored at 7 °C. All these defects in milk samples may be related to changes in the flora due to the storage temperature. In fact, Griffiths and Phillips [31] reported that the refrigeration temperatures spoilage was predominantly due to *Pseudomonas* species growth than that of storage temperature. There was a rather poor relationship among sensory and microbiological data with respect to total coliform, psychotropic and standard plate counts. As formerly recognized [20,32], the spoilage of a specific food substrate is the result of the growth of specific spoilage organisms and not of total microflora. Furthermore, these outcomes exposed that there was a low relationship between microbiological and volatile compound data, and as for a large number of volatile compounds, there was no variation in concentration with the growth of microbial populations, particularly at 4 °C (Figures 2–10). Pondering the relationship between sensory and volatile compounds, there was a small rise in acetone, butanone, pentanal, heptanal, ethanol and pentane concentrations with storage time, at 4 °C. The accumulation of these compounds seems to link well with reducing flavor scores of milk (Figures 2–10).

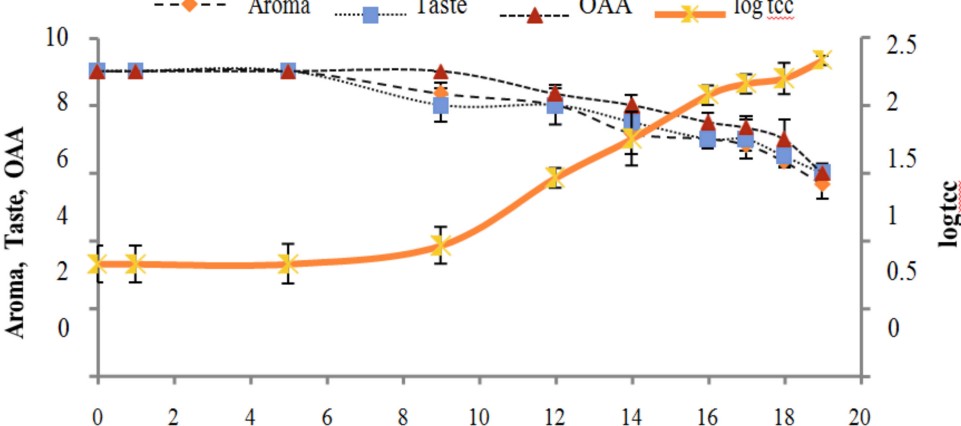

**Figure 2.** Relationship of total coliform count Log CFU/mL growth trends with sensory scores of pasteurized milk for spoilage detection stored at 4 °C.

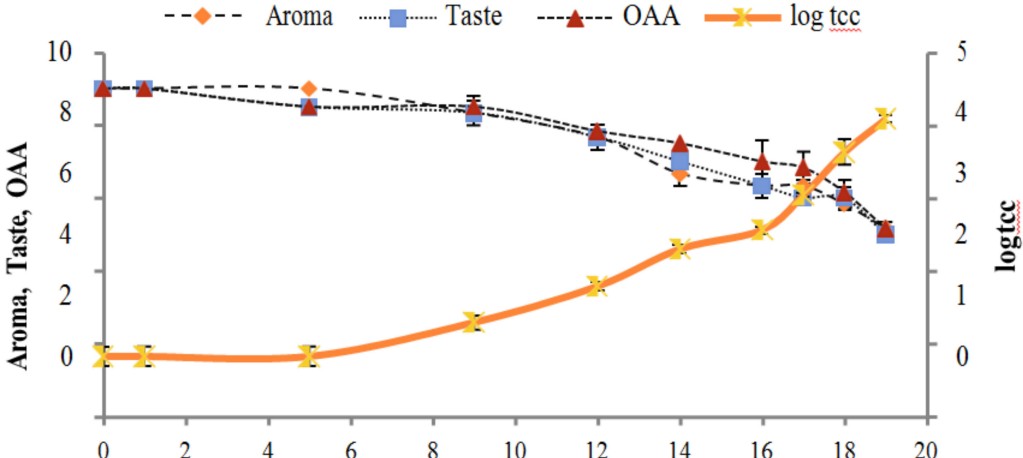

**Figure 3.** Relationship of total coliform count log cfu/mL growth trends with sensory scores of pasteurized milk for spoilage detection stored at 7 °C.

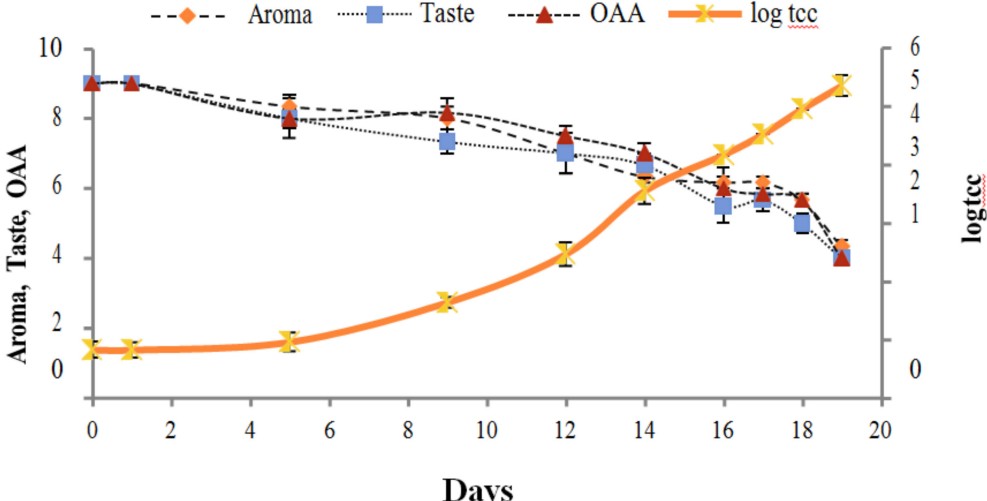

**Figure 4.** Relationship of total coliform count log cfu/mL growth trends with sensory scores of pasteurized milk for spoilage detection stored at 10 °C.

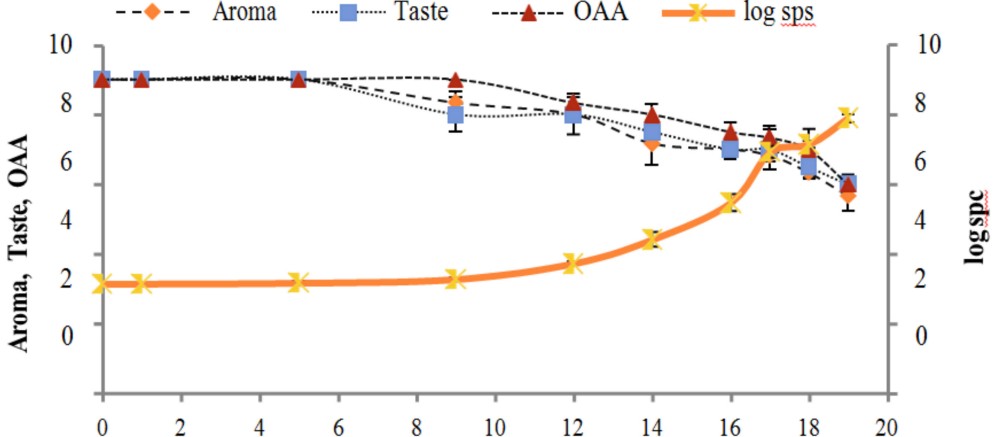

**Figure 5.** Relationship of standard plate count log cfu/mL growth trends with sensory scores of pasteurized milk for spoilage detection stored at 4 °C.

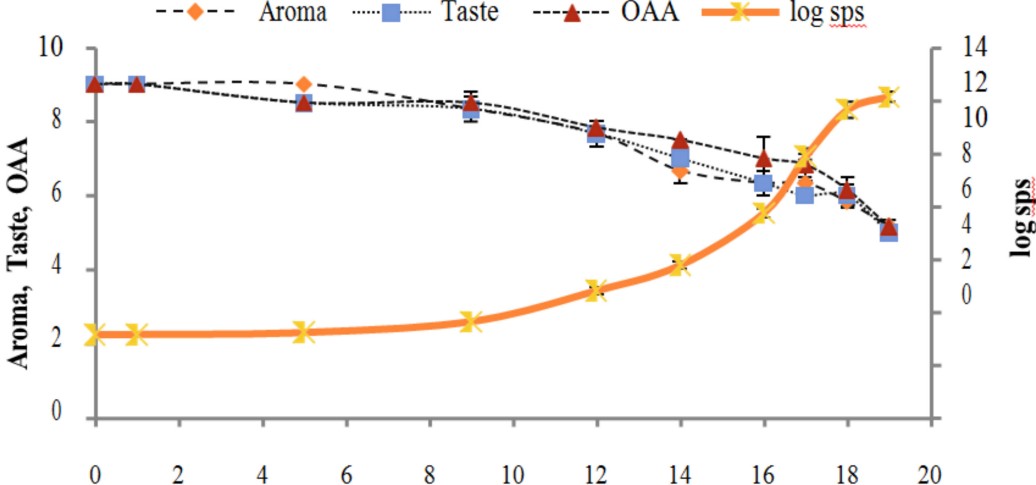

**Figure 6.** Relationship of standard plate count log cfu/mL growth trends with sensory scores of pasteurized milk for spoilage detection stored at 7 °C.

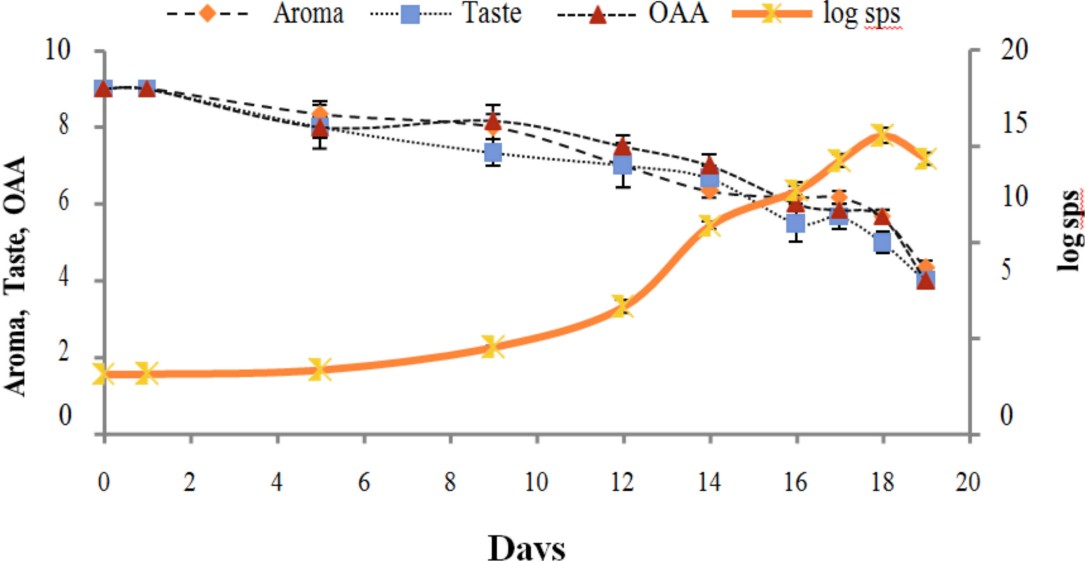

**Figure 7.** Relationship of standard plate count log cfu/mL growth trends with sensory scores of pasteurized milk for spoilage detection stored at 10 °C.

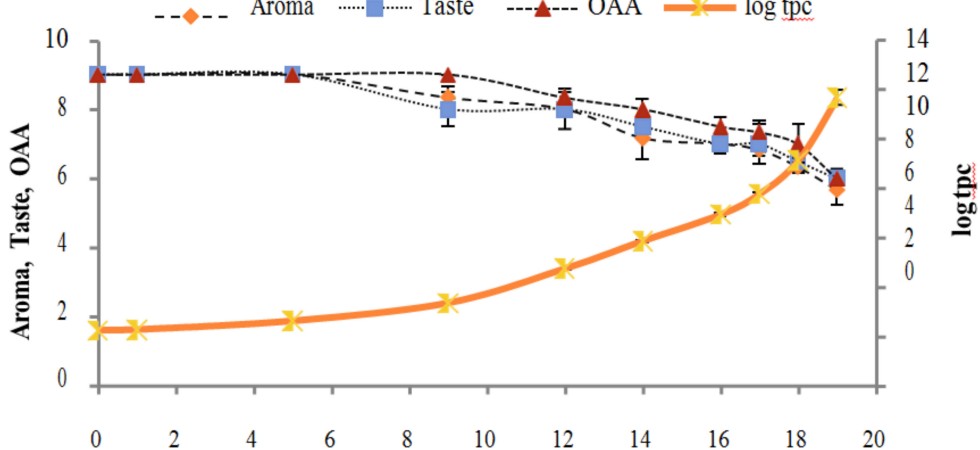

**Figure 8.** Relationship of total psychrotrophs count log cfu/mL growth trends with sensory scores of pasteurized milk for spoilage detection stored at 4 °C.

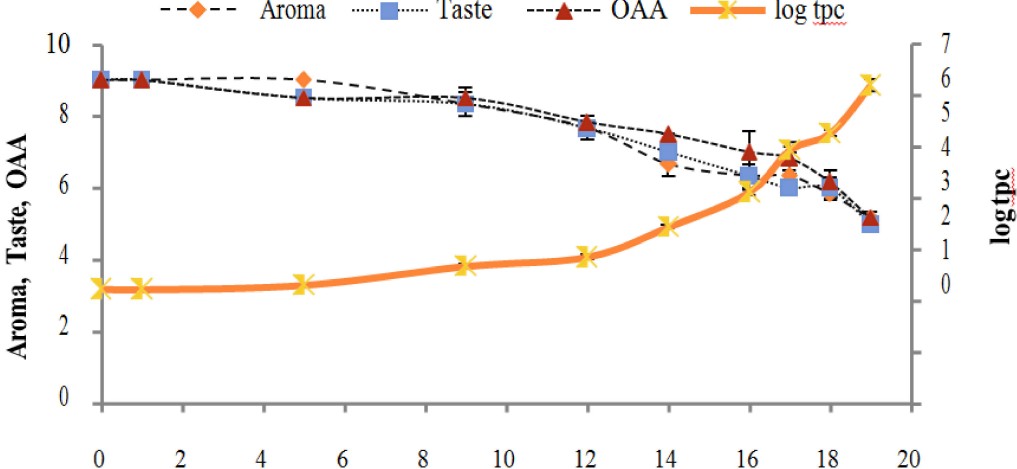

**Figure 9.** Relationship of total psychrotrophs count log cfu/mL growth trends with sensory scores of pasteurized milk for spoilage detection stored at 7 °C.

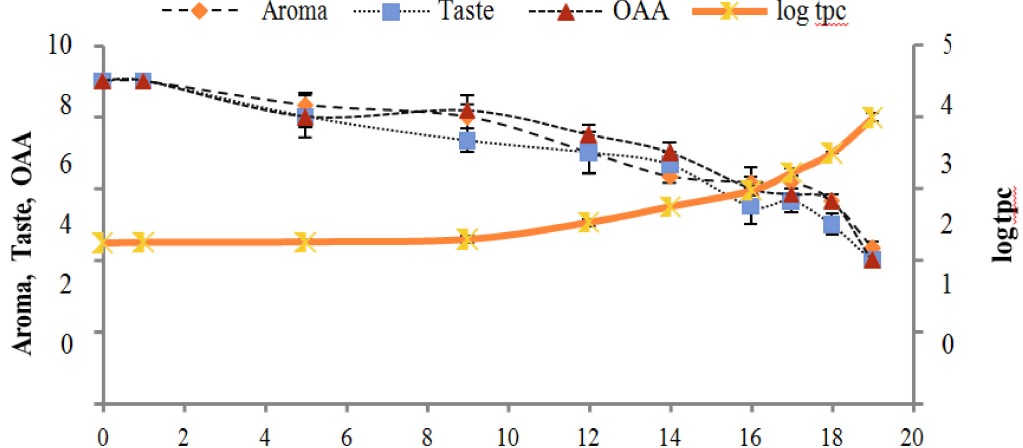

**Figure 10.** Relationship of total psychrotrophs count log cfu/mL growth trends with sensory scores of pasteurized milk for spoilage detection stored at 10 °C.

## 4. Conclusions

Overall, the HS-SPME/GC-FID methodology developed in this study was revealed to be very sensitive and capable to be applied in volatile compound quantification in pasteurized milk produced during the storage period at different temperatures as an indicator of spoilage. Overall, HS-SPME/GC-FID was able to extract, separate and identify the volatile compounds from the milk sample; but it still had flaws regarding information on the key aroma compounds. Further studies should be performed to separate and identify the key aroma compounds of pasteurized milk samples in the near future.

**Author Contributions:** A.R. and I.J. drafted this project; B.R. and S.S. supported and facilitated the part of research project in Washington State University, USA; M.A. and M.A.A. prepared methodology; M.A. applied the software on this research; M.I. validated, gave formal analysis, and data correction; T.A.G. and M.I.A. wrote original draft; B.S., C.F.R., J.S.-R., and N.M. prepared draft, writing—review and editing, visualization, and project administration, and funding provided by International Research Support Initiative Program (IRSIP), Higher Education Commission, Islamabad, Pakistan.

**Funding:** This research received no external funding.

**Acknowledgments:** The authors are grateful to the management team at Walmart Superstore, Pullman, Seattle, USA, which provided freshly processed pasteurized milk from Darigold, Inc. They acknowledge Professor Barbara Rasco and Assistant Professor Dr. Shyam S. Sablani (School of Food Sciences, Washington State University, USA) for their valuable suggestions and laboratory supply. They are also grateful to the Higher Education Commission,

Islamabad, Pakistan, for granting the International Research Support Initiative Program (IRSIP) as total funds for this project. Natália Martins would like to thank the Portuguese Foundation for Science and Technology (FCT-Portugal) for the Strategic project ref. UID/BIM/04293/2013 and "NORTE2020—Northern Regional Operational Program" (NORTE-01-0145-FEDER-000012), and Célia F. Rodrigues for the UID/EQU/00511/2019 Project—Laboratory of Process Engineering, Environment, Biotechnology and Energy—LEPABE financed by national funds through FCT/MCTES (PIDDAC).

**Conflicts of Interest:** The authors declare no conflict of interest.

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
