# Peer review of "Measurement of Off-Flavoring Volatile Compounds and Microbial Load as a Probable Marker for Keeping Quality of Pasteurized Milk"

_applsci, doi:10.3390/app9050959_

Round 1

Reviewer 1 Report

In my opinion, the methodical part of the work needs to be supplemented. The methodology for the determination of volatile compounds using HS-SPME needs to be supplemented. What standard was used to quantify HS-SPME? Was it an external or internal standard?

According to what standards, microbiological tests were carried out?

Author Response

In my opinion, the methodical part of the work needs to be supplemented. The methodology for the determination of volatile compounds using HS-SPME needs to be supplemented. What standard was used to quantify HS-SPME? Was it an external or internal standard?

Answer: According to your suggestion and recommendation the methodology for the determination of volatile compounds using HS-SPME has been supplemented. An internal standard was used to quantify HS-SPME now it has been explained in methodology part

According to what standards, microbiological tests were carried out?

Answer: The microbiological tests were performed according to standard methods of American Association of Cereal Chemists (AACC), method numbers have been mentioned in the methodology.

Reviewer 2 Report

The authors presented a paper on the measurment of off-flavors in milk and correlated this to microbial analysis. This paper however needs further improvements prior to publishing.

In their introduction the authors state that is difficult to estimate the keeping qualtity of milk and present their method to help solve this problem. I'm missing however a conclusion on how this method can predict the keeping quality of the milk. Currently it identifies the compounds which cause the off flavor once the milk is already scoring bad in an taste panel evaluation. As such the method has no added value for industy.

From the experimental part it is not clear

How the authors quantified the off-flavors

How many samples were analysed per time/temp point

I would suggest to move the section on optimization of the SPME method prior to the SPME results

Why did the authors not test the CAR/PDM/DVB fiber which combines the properties of the CAR/PDMS and PDMS/DVB fiber?

I find it strange that the authors conclude that their results correlate well with those of Aardt et al and Cladman et al while these two references talk about the impact of the package of the milk while the authors studied impact of time and temperature

It is not clear how the aroma is presented in figuers 2 - 10. In these figures aroma values drop with storage time while from the results in tables1-3 we can cleary see that more compounds are detected at higher concentration during storage

It suggested to see if by processing the obtained results for aroma, taste panels and microbial test with a pattern recognition technique more correlation with the different test and more conclusive results on the main aroma can be made.

The quality of figure 1 needs to be improved

Several references to similar work are missing

Lloyd, M. A., Drake, M. A., & Gerard, P. D. (2009). Flavor variability and flavor stability of U.S.-produced whole milk powder. Journal of Food Science, 74(7), 334–343. https://doi.org/10.1111/j.1750-3841.2009.01299.x

Vazquez-Landaverde, P. a, Velazquez, G., Torres, J. a, & Qian, M. C. (2005). Quantitative determination of thermally derived off-flavor compounds in milk using solid-phase microextraction and gas chromatography. Journal of Dairy Science, 88(11), 3764–3772. https://doi.org/10.3168/jds.S0022-0302(05)73062-9

Jensen, S., Jansson, T., Eggers, N., Clausen, M. R., Larsen, L. B., Jensen, H. B., … Bertram, H. C. (2015). Storage-induced changes in the sensory characteristics and volatiles of conventional and lactose-hydrolyzed UHT processed milk. European Food Research and Technology, 240(6), 1247–1257. https://doi.org/10.1007/s00217-015-2427-9

Author Response

The authors presented a paper on the measurment of off-flavors in milk and correlated this to microbial analysis. This paper however needs further improvements prior to publishing.

In their introduction the authors state that is difficult to estimate the keeping qualtity of milk and present their method to help solve this problem. I'm missing however a conclusion on how this method can predict the keeping quality of the milk. Currently it identifies the compounds which cause the off flavor once the milk is already scoring bad in an taste panel evaluation. As such the method has no added value for industy.

Answer: Basically, the taste panel of sensory evaluation was carried out to just check its compatibility with the results of microbial study and off flavoring determination for keeping quality of milk. The main work is the method to measure the off flavoring compounds to solve the problem of keeping quality of pasteurized milk. So, the method of HS-SPME is useful to detect off flavoring compounds that give it bad score.  

From the experimental part it is not clear

How the authors quantified the off-flavors.

Answer: The experimental part has been revised and supplemented to explain how the off-flavors were quantified

How many samples were analyzed per time/temp point

Answer: 30 samples in three replicates were analyzed per time/temperature point

I would suggest to move the section on optimization of the SPME method prior to the SPME results

Answer: It has been moved to the place as per your suggestion.

Why did the authors not test the CAR/PDM/DVB fiber which combines the properties of the CAR/PDMS and PDMS/DVB fiber?

Answer: CAR/PDM/DVB fiber has been tested for another project of cheese.

I find it strange that the authors conclude that their results correlate well with those of Aardt et al and Cladman et al while these two references talk about the impact of the package of the milk while the authors studied impact of time and temperature.

Answer: These references has been changed as per your suggestion

It is not clear how the aroma is presented in figuers 2 - 10. In these figures aroma values drop with storage time while from the results in tables1-3 we can cleary see that more compounds are detected at higher concentration during storage

Answer: Basically here aroma represents the good flavor of milk. Tables 1-3 indicate that the concentration of off flavoring compounds was increased and as the result of these bad flavoring compounds the actual flavor/aroma values (figures 2-10) of milk were dropped with storage. 

It suggested to see if by processing the obtained results for aroma, taste panels and microbial test with a pattern recognition technique more correlation with the different test and more conclusive results on the main aroma can be made.

The quality of figure 1 needs to be improved

Answer: Quality has been improved

Several references to similar work are missing

Lloyd, M. A., Drake, M. A., & Gerard, P. D. (2009). Flavor variability and flavor stability of U.S.-produced whole milk powder. Journal of Food Science, 74(7), 334–343. https://doi.org/10.1111/j.1750-3841.2009.01299.x

Vazquez-Landaverde, P. A., Velazquez, G., Torres, J. A., & Qian, M. C. (2005). Quantitative determination of thermally derived off-flavor compounds in milk using solid-phase microextraction and gas chromatography. Journal of Dairy Science, 88(11), 3764–3772. https://doi.org/10.3168/jds.S0022-0302(05)73062-9

Jensen, S., Jansson, T., Eggers, N., Clausen, M. R., Larsen, L. B., Jensen, H. B., … Bertram, H. C. (2015). Storage-induced changes in the sensory characteristics and volatiles of conventional and lactose-hydrolyzed UHT processed milk. European Food Research and Technology, 240(6), 1247–1257. https://doi.org/10.1007/s00217-015-2427-9

Answer: It has been incorporated the missing references to similar work

Round 2

Reviewer 2 Report

-